# The Utilization of Physiologically Active Molecular Components of Grape Seeds and Grape Marc

**DOI:** 10.3390/ijms231911165

**Published:** 2022-09-22

**Authors:** Imre Hegedüs, Kitti Andreidesz, József L. Szentpéteri, Zoltán Kaleta, László Szabó, Krisztián Szigeti, Balázs Gulyás, Parasuraman Padmanabhan, Ferenc Budan, Domokos Máthé

**Affiliations:** 1Department of Biophysics and Radiation Biology, Semmelweis University, Üllői út 26, H-1085 Budapest, Hungary; 2Department of Biochemistry and Medical Chemistry, University of Pécs Medical School, H-7624 Pécs, Hungary; 3Institute of Transdisciplinary Discoveries, Medical School, University of Pecs, 48-as tér 1, H-7622 Pécs, Hungary; 4Higher Education and Industrial Cooperation Centre, University of Miskolc, H-3515 Miskolc, Hungary; 5PROGRESSIO Engineering Bureau Ltd., Muhar u. 54, H-1028 Budapest, Hungary; 6Department of Public Health Medicine, Medical School, University of Pécs, H-7624 Pécs, Hungary; 7Lee Kong Chian School of Medicine, Nanyang Technological University, 50 Nanyang Avenue, Singapore 639798, Singapore; 8Institute of Physiology, Medical School, University of Pécs, H-7624 Pécs, Hungary; 9In Vivo Imaging Advanced Core Facility, Hungarian Centre of Excellence for Molecular Medicine, H-1094 Budapest, Hungary

**Keywords:** grape, polyphenols, flavonoids, anticarcinogen, vasoprotective, antidiabetic, antioxidant, anti-inflammatory

## Abstract

Nutritional interventions may highly contribute to the maintenance or restoration of human health. Grapes (*Vitis vinifera*) are one of the oldest known beneficial nutritional components of the human diet. Their high polyphenol content has been proven to enhance human health beyond doubt in statistics-based public health studies, especially in the prevention of cardiovascular disease and cancer. The current review concentrates on presenting and classifying polyphenol bioactive molecules (resveratrol, quercetin, catechin/epicatechin, etc.) available in high quantities in *Vitis vinifera* grapes or their byproducts. The molecular pathways and cellular signaling cascades involved in the effects of these polyphenol molecules are also presented in this review, which summarizes currently available in vitro and in vivo experimental literature data on their biological activities mostly in easily accessible tabular form. New molecules for different therapeutic purposes can also be synthesized based on existing polyphenol compound classes available in high quantities in grape, wine, and grape marc. Therefore an overview of these molecular structures is provided. Novel possibilities as dendrimer nanobioconjugates are reviewed, too. Currently available in vitro and in vivo experimental literature data on polyphenol biological activities are presented in easily accessible tabular form. The scope of the review details the antidiabetic, anticarcinogenic, antiviral, vasoprotective, and neuroprotective roles of grape-origin flavonoids. The novelty of the study lies in the description of the processing of agricultural by-products (grape seeds and skins) of industrial relevance, and the detailed description of the molecular mechanisms of action. In addition, the review of the clinical therapeutic applications of polyphenols is unique as no summary study has yet been done.

## 1. Introduction

*Vitis vinifera* grapes are extremely rich in bioactive components [1]. Grape marc is a mixture of grape seeds and skins, which remain as a by-product of the wine production process, making up 20–25% of the grape’s weight [2]. Grape seeds contain fats, proteins, carbohydrates, and 5–8% polyphenols. The grape seed is rich in extractable phenolic antioxidants such as phenolic acids, flavonoids, proanthocyanidins, and resveratrol, and the grape skin is abundant in anthocyanins [3]. Grape marc also contains a large amount of lipids, proteins, indigestible fibers, and minerals [1,2].

Around 1.000 kg of grapes is used to produce 750 L of wine. By way of comparison of start- and end-product masses, this means that about 60% of the grape harvest mass will become agricultural waste [4]. As an example, in 2017, Chinese grape production was 13,083,000 tons and South African grape production was 2,032,582 tons [5]. Hence, there is a huge untapped potential in the use and extraction of active substances from grape seeds, skins, and pomace.

### Polyphenols

Polyphenols are so-called secondary metabolites of plants, biologically active compounds in order to enhance plants adaptation to environmental conditions, for example balancing oxidative stress [6]. Polyphenols are plants’ active substances consisting of more than one phenolic group. In food, more than 15 classes of polyphenols can be found [7]. The polyphenols are largely flavonoids that can be further subdivided into 13 subclasses where more than 8000 components have been described. Flavonoids are the largest and most-studied group of phenols. Their seven main subclasses are flavones, flavonones, flavonols, isoflavones, anthocyanidins/anthocyanins, flavanols (or catechins and procyanidins), and chalcones [7]. Another group of flavonoids not included in this list are proanthocyanidins, also known as procyanodins, condensed tannins, or oligomeric procyanidins [7]. High molecular weight (from 500 D up to 20,000 D) polyphenols are plant tannins [8]. Polyphenols can generally be subdivided into hydrolyzable tannins (tannic acid esters with glucose or other sugars) [9], phenylpropanoids (lignins, flavonoids) [10,11,12], and condensed tannins [13]. Polyphenol compounds, especially procyanidins, contribute to the bitter and astringent taste of juices shaping the aroma of wines [14]. The coloring agents from the grape skins are considered “generally recognized as safe” (GRAS) and are utilized as food colorants [14].

In grapes, flavonoids are mainly found in the seeds, fruit skins, and stems. Between 60 and 70% of the total recoverable polyphenols in grapes are in the seed, which accounts for 5–8% of the weight of the seed [15]. Hundreds of polyphenolic compounds are present in wine, which influence the taste, color, and flavor of the wine [14]. The extractable phenolic antioxidants account for 10–11% of the dry weight of the grape marc. The polyphenolic composition of marc is varietal. Red grapes are richer in proanthocyanidins, while in white grapes they are scarcely present. The composition of polyphenols depends on the grape variety, the weather, the place of cultivation, and the maturity of the grapes [16]. The largest and best-known constituents of polyphenols are flavonoids [17] (Figure 1). The vast majority of polyphenols in grape seeds are flavonoids [18]. The classification of polyphenols and the characteristic functions of each molecular class are summarized in Table 1.

The main polyphenolic constituents in grape seeds are catechins (catechin, epicatechin, procyanidin [19]). Except for epicatechin, they are found in the outer, soft layer of the grape seed. The most physiologically important compounds of polyphenols isolated from grape seeds are summarized in Table 2.

## 2. Review Methodology

Our aim was to prepare a scoping review to demonstrate that there is a significant amount of active substances in grapes, mainly in the seeds and pomace, which in many cases become waste. We also provide an overview of the wide range of physiological effects of these available active substances. Therefore, the extraction and use of these molecules as food supplements or possibly as novel pharmaceutical concepts such as dendrimer nano-bioconjugates could have a significant health-enhancing and disease-preventive effect on the population.

To access relevant articles the Web of Science and PubMed databases were used, augmented with the Google Scholar search engine. The “polyphenols” and “grape” keywords were applied, and 5981 results have been found on Web of Science, 3690 results on Pubmed, and more than 131,000 articles, dissertations, and scientific reports in Google Scholar.

Later on, these keywords were supplemented with keywords for the most typical ingredients (“resveratrol”, “quercetin”, “tannin”, “anthocyanin”) or the most prominent physiological effects (“antioxidant”, “free radical scavenger”, “Anti-atherosclerotic”, “cardioprotective”, “nervous system”, “anti-inflammatory”, “anti-cancer”, “signal transduction”, “endothel”, “blood vessel”, “diabetes”, “cell cycle”, “bioactive”, “in vivo”, “clinical”, “preventive”, “therapeutic”) and these results were compiled. The authors also discuss the extraction of polyphenols and their technological potential as food additives in Appendix A, to ease the overview. Finally, some of the pharmaceutical applications of polyphenols are listed using nanobioconjugates such as dendrimers.

## 3. Basic Physical and Chemical Properties of Polyphenols

### 3.1. Physical Properties

The most important physical properties of the main monomeric components of polyphenols, such as catechol, epicatechin (EC), and epicatechin-(3-O)-gallate (EGC) monomers are summarized in Table 3. Properties relevant for the separation of the components:

Molecular weight: Based on the differences in molecular weight, it is possible to separate fractions by gel chromatography or membrane filtration.

Solubility: Several polyphenols are water-soluble, and many are lipid-soluble. In general, catechins are fat soluble and procyanidins are water-soluble. This allows their relatively easy separation by extraction.

### 3.2. Chemical and Biochemical Properties of Polyphenols

Hydrogen donor: Polyphenols have numerous hydroxyl groups, acting as hydrogen donor antioxidants, and scavenging singlet oxygen. Therefore, they can be classified as reducing agents. They form chelates with metals. They bind free radicals and stop radical chain reactions [20].

Stability: The antioxidants that can be extracted from grape seeds are very sensitive to oxygen, light, acidic, or alkaline environments, and variably sensitive to heat [21].

Polyphenol-protein interactions: The polyphenols in grape seed extract can form strong, specific bonds with protein binding units (e.g., proline-rich proteins). This binding is used in the extraction of polyphenols and the extraction of plant proteins (gluten removal). They can also inhibit or activate enzymes in grapes that protect the fruit from microbial attack. The interactions between polyphenols and proteins can be covalent, ionic, hydrogen-bonded, or hydrophobic. Many proteins can precipitate polyphenols [22].

### 3.3. Analysis of Polyphenols

Several simple and inexpensive methods for the analytical determination of antioxidants have been developed (Table 4). Antioxidant activity can be determined simply by the FRAP method (FeCl_3_ and using triazine) [23], and total antioxidant content can be determined by András Boór’s determination of 2,4,6-Tris(2-pyridyl)-s-triazine [23], total polyphenol content can be determined by Folin Ciocalteu reagent [23]. Free radical scavenging activity can be measured using 1,1-Diphenyl-2-picrylhydrazine [23]. The anthocyanin content in hydrochloric acid ethanol can be determined spectrophotometrically (550 nm) [24], and leucoanthocyanins can be determined spectrophotometrically after heating with a hydrochloric acid-butanol mixture containing ferrous sulfate (II) in a 40:60 ratio [24]. Determination of catechol content in alcohol diluted solution reacted with sulfuric acid vanillin at 500 nm spectrophotometrically is simple [25], and resveratrol content can be determined directly in HPC by the Kállay-Török method [26] (Table 4).

## 4. The Beneficial Effects of Polyphenols on Health and Its Molecular Mechanisms

Polyphenols possess multifarious beneficial in vivo and clinical health effects, and their details are out of the scope of this review article. In summary, the evident antioxidant and reactive oxygen species inhibitory effects of polyphenols promote the following intracellular signal transduction and regulation process: they downregulate inflammatory proteins (for example IL-1, IL-6, TNFα, mTOR, Nuclear Factor-κB) and also oncogenes (for example C-MYC, RAS, NOTCH) and by the time upregulate tumor suppressor genes (for example SIRT, PTEN, P53) expression through epigenetic factors, such as histone acetylation, DNA methylation, and microRNA expression) [29,30,31,32]. These molecular effects and intracellular signals lead ultimately to the well-known clinically proven cardiovascular protective, anticarcinogen, antidiabetic, antimicrobial, etc. effects.

### 4.1. Antioxidant and Free Radical Scavenging Activity

The main physiological effects of grapeseed oil are its antioxidant properties and its ability to bind free radicals. The total polyphenol fraction in grape seed is characterized by these properties. The antioxidant effect of polyphenols is 20 times stronger than vitamin E and 50 times stronger than vitamin C. They protect LDL and cholesterol from oxidation and prevent platelet aggregation, thus they prevent coronary heart disease and maintain vascular integrity. Also polyphenols are capable of protecting the skin from sunburn [33,34], rejuvenating the skin, and preserving the flexibility and elasticity of joints, blood vessels, and tissues. Flavonols from grape seeds reduce alcohol-induced lipid peroxidation (lipofuscin formation) and thus protect the brain from the damaging effects of alcohol [35] Grape seed extract also protects against age-related DNA damage in the central nervous system, preventing DNA oxidation and the formation of DNA-protein bonds [36]. For a detailed mechanism of action broken down into molecules, see Table 5.

### 4.2. Anti-Atherosclerosis and Cardioprotective Effects

Grape seed extract does possess a cardioprotective effect, for example, it reduces the likelihood of heart attack [52]. Furthermore, it is also beneficial for numerous other cardiovascular diseases. Quercetin inhibits the oxidation of LDL and cholesterol and the clumping of platelets. Other beneficial effects are exerted by resveratrol, epicatechin, epigallocatechin gallate (EGCG), epicatechin gallate (ECG), genistein, and daidzein, namely, they protect against atherosclerosis and alleviate arrhythmias of the heart [53]. Polyphenols reduce the risk of cardiovascular disease, for example, they decrease the risk of coronary heart disease. They act by dilating the blood vessel walls, thereby reducing blood pressure [54,55,56]. They also reduce women’s blood pressure by regulating estrogen hormones [57] (Table 6).

### 4.3. Neuroprotective Effects

Through the regulation of several enzymes, resveratrol protects nerve tissue from fibrosis, cell death in the case of prolonged damage, protects dopaminergic neurons, and prevents beta-amyloid deposition in prolonged inflammatory processes. EGCG, ECG, and myricetin inhibit the growth of tumors in the nervous system through the regulation of intracellular enzymes. Catechin and quercetin inhibit the progression of programmed cell death in the event of injury. For the molecular mechanisms of action of each fraction, see Table 7.

### 4.4. Anti-Inflammatory Effect

Polyphenols inhibit the action of several histamine-releasing enzymes and (among others) therefore have anti-inflammatory and anti-allergic effects. Grape seed extract promotes the healing of autoimmune rheumatoid arthritis [70]. Procyanidins, EGCG and EGC reduce the inflammatory activation of peripheral monocyte cells. Procyanidins also have anti-ulcer effects [71]. Resveratrol reduces inflammation in articular cartilage cells and prostate cells. Quercetin acts in vessel walls and macrophages, and anthocyanins in small blood vessels. They inhibit lipid peroxidation and DNA fragmentation in the liver and brain. For the molecular mechanisms of action of each fraction, see Table 8.

### 4.5. Mutation Reduction and Anti-Cancer Effect

Different types of procyanidins inhibit the proliferation of cancer cells and thereby inhibit metastasis formation, for example, prostate tumors [82,83]. Proanthocyanidins from grape seed extract also induce programmed cell death (apoptosis) in metastases of advanced breast cancer [84]. Furthermore, they inhibit colon cancer growth too [85]. For molecular mechanisms of action of each fraction, see Table 9.

### 4.6. Influencing Signal Transduction

In many cases, polyphenols in grape seeds affect interstitial, extra- and intracellular information transduction mechanisms through the cell membrane effects. Proanthocyanidins accelerate programmed cell death in cancer cells, quercetin enhances the functionality of primary cortical neurons through signal transduction. Resveratrol reduces inflammatory overactivation of monocytes via signaling and inhibits cardiac fibrosis (see Table 10).

### 4.7. Effects on the Vascular Wall and Choroidal Cells

EGCG and quercetin reduce programmed cell death in the cells that build up the vascular wall. In calf vascular endothelial cell culture, Cy3G increases cell lifespan through cyclic guanosine monophosphate (cGMP) and nitric oxide (NO) regulation. In addition, EGCG increases vasodilator effects in calf aortic vascular endothelial cells. Catechins reduce the vascularization-inducing effect of angiogenin-like proteins in chickens. Proanthocyanidins reduce inflammation-induced cell damage in human choroidal cells. Proanthocyanidins and flavan-3-ols reduce the degradation of enzymes responsible for the relaxation of blood vessels (see Table 11).

### 4.8. Effects on Diabetes

EGCG, ECG, and (-)-EGC polarize the intestinal epithelial cells responsible for sugar uptake and inhibit sugar uptake in the rabbit’s small intestine. Quercetin lowers blood glucose levels, tannins and anthocyanin inhibit the alpha-amylase enzymes responsible for glycogen degradation (which increases blood glucose levels) and the alpha-glucosidase enzymes in the intestinal wall responsible for sugar absorption (see Table 12).

### 4.9. Effects on the Cell Cycle

Resveratrol stops HepG2 liver cancer cells from dividing by stimulating the expression of the P21 protein, which inhibits the CDK cyclin-dependent kinase enzyme. In human skin tumor and human colon cancer cells, it inhibits the formation of complexes of cyclins involved in cell cycle regulation, reduces the phosphorylated form of the pRb enzyme responsible for initiating DNA synthesis during cell division and leads to cell cycle arrest in the G0/G1 phase through inhibition of the expression of the transcription factors E2F (1–5) and their heterodimeric partners DP1, DP2. Proanthocyanidins inhibit the expression of cell cycle regulators cyclin B1, D1, A1, and β-catenin, which accumulate in cancer cells and are responsible for inappropriate gene activity. They arrest the cell cycle in the G1 phase, reducing the expression of cyclins in human melanoma cells. In human skin cancer cells, they promote cell cycle arrest in the G1 phase, inhibit the function of cyclins and cyclin-dependent kinases (CDK), and promote the expression of CDK inhibitors (see Table 13).

### 4.10. Other Impacts

#### 4.10.1. Anti-Caries Effect

In the case of caries of the tooth root, proanthocyanidin-containing grape seed extract induces the re-crystallisation of the tooth enamel [118].

#### 4.10.2. Antihyperlipidemic Effect

Grape seed extract has been shown in clinical trials to increase satiety, reduce energy intake from food intake, and increase fat breakdown in vitro [119]. Grape seed extract inhibits the enzymes involved in fat metabolism (pancreatic lipase, lipoprotein lipase), thus preventing the accumulation of fat in adipose tissue [120]. Mice fed grape seed extract have reduced tissue fat levels but not influenced body weight [121]. Polyphenols isolated from grape seeds and red wine inhibit intracellular cholesterol synthesis, and thereby reduce blood cholesterol level [122].

#### 4.10.3. Antibacterial and Antifungal Effect

The antibacterial activity of polyphenols covers both Gram-positive and Gram-negative bacteria. It also enhances antifungal effects, for example in the case of *Candida albicans* infection (candidiasis) it increased the efficacy of medicines [123].

#### 4.10.4. Anti-HIV Effect

Proanthocyanidins inhibit the expression of HIV-secreting coreceptors in normal peripheral mononuclear cells [124].

#### 4.10.5. Sensory Effect

Proanthocyanidins and resveratrol enhance the expression of vascular endothelial growth factor (VEGF) in pigment cell culture [124,125] and animal model (hamsters) [126].

#### 4.10.6. Hepatoprotective Effect

Novel proanthocyanidins IH636 increase the expression of the mitochondrial signal transduction enzyme BCL-xL and attenuate acetaminophen-induced hepatic DNA damage and programmed and necrotic destruction of liver cells in an engineered mutant (ICR) mouse strain. In rat liver, daidzein improves the growth of d-galactosamine-induced malondialdehyde-protein adducts and cytoplasmic superoxide dismutase (SOD) activity. In rats, genistein reduces experimental liver damage caused by CCl_4_ by preventing lipid peroxidation and enhancing the antioxidant system (see Table 14).

### 4.11. Anti-SARS-CoV-2 Effect

Beneficial effects of tannins against “cytokine storm” of severe acute respiratory syndrome coronavirus 2 (SARS-CoV-2) are also effectively investigated recently. It has been discussed that tannins have both preventive and therapeutic potential against SARS-CoV-2 infections [13,128].

### 4.12. Risks Associated with Polyphenols

To date, no adverse effects of polyphenols have been reported at low concentrations [129]. The adverse effects of polyphenols are observed at high doses. Adverse effects on the body may include carcinogenesis/genotoxicity, thyroid damage, the estrogenic activity of isoflavones, dietary effects, and drug interactions [130].

Grape seed extract may contain isoflavones at a concentration of 50 mg/dose or proanthocyanidins at a concentration of 100–300 mg/dose. In rats and mice, 0.5–2.0 g of proanthocyanidin extracted from grape seeds per kg body weight does not cause acute toxicity [131]. In rats, 60 g/kg of ellagitannin does not cause acute toxicity [132]. However, chronic kidney damage has been observed in rats when given high doses (2% or 4% per kg body weight) of quercetin in their diet [133]. A reduction in life expectancy has been observed even when 0.1% quercetin was added to the diet of mice [134].

Some polyphenols are carcinogenic and genotoxic at higher concentrations [135]. For example, caffeic acid at 2% in the diet induces gastric and renal tumors in mice and rats [136]. High levels of quercetin can cause tumor formation [137]. Catechins in high doses promote the division of tumor cells in the colon, but quercetin reduces the division of tumor cells [138,139].

At high doses, proanthocyanidins (also known as procyanidin oligomers) can cause liver damage, hemophilia, tumors of the female reproductive organs and inflammation of the intestine [140].

Some flavonoids were reported to inhibit thyroid hormone synthesis [141,142]. Among the isoflavones (flavanols), a decrease in thyroid hormone levels has been observed in rats at high doses of genistein [143], as well as disruption of some female hormones [144]. However, isoflavones are present in grape seeds only in negligible amounts.

Polyphenols are also known to inhibit nutrient absorption. For example, they impair iron absorption [145], but vitamin C, which may be present in nutrient sources alongside polyphenols, promotes iron absorption [146]. In addition, proanthocyanidins (condensed tannins) may also have an inhibitory effect on nutrient absorption as they can interact with proteins and inhibit the function of several enzymes. However, these properties are only seen at extremely high doses (10 g/kg body weight in the diet) and not at lower doses [147]. Diets high in tannins may show a protein absorption-reducing effect [148]. It should be noted that such high polyphenol content is not found in Western diets.

Finally, polyphenols can affect the bioavailability and mechanism of action of drug substances. Some drugs, such as benzodiazepines and terfenadine, may have a threefold increase in bioavailability in the presence of polyphenols (due to CYP3A4 inhibition) [149].

## 5. In Vivo Investigations of Grape Seed Extract and Its Components

The number of in vivo investigations using a wider range of immunohistochemistry to examine various molecular mechanisms of action is growing exponentially (Table 15).

By using MTT assay, flow cytometry, and immunoblot analysis, lipophilic grape seed proanthocyanidin (LGSP) was assessed for its anti-prostate cancer activity against the PC3 cell line in vitro. A mouse xenograft model generated from PC3 was used to test LGSP’s anti-prostate cancer impact in vivo. In tumor tissues, immunostaining tests for Ki67 and cleaved caspase 3 were carried out. By triggering apoptosis, LGSP had a potent inhibitory effect on PC3 cell proliferation [150].

Treatment with LGSP caused G1 phase cell cycle arrest in PC3 cells, which was further validated by increased expression of the tumor suppressor p21 and p27 and decreased expression of cyclin D1 and CDK 4. Additionally, it was demonstrated that LGSP-induced apoptosis is caspase-dependent by the activation of cleaved fragments of caspases 3, caspase 9, as well as PARP. LGSP boosted the release of cytochrome c in the cytoplasm upstream of the caspase cascade. In PC3 cells following LGSP administration, the Bcl-2/Bax ratio likewise fell [150].

Considering tumor research, LGSP inhibited the proliferation of PC3-derived mouse xenografts and induced apoptosis [150].

Nude mice with a human liver cancer cell (HepG2)-derived xenografts were treated with grape seed proanthocyanidins (GSPs). According to the findings, GSPs triggered autophagy, and inhibiting autophagy caused an increase in apoptosis in HepG2 cells. Since stimulating the phosphorylation of mitogen-activated protein kinase (MAPK) pathway-associated proteins, p-JNK, p-ERK, and p-p38 MAPK, and decreasing the expression of survivin, GSPs at 100 mg/kg and 200 mg/kg significantly inhibited the proliferation of HepG2 cells in nude mice without manifesting toxicity or autophagy [151].

Grape seed procyanidin reverses the change in pulmonary hemodynamics in the cigarette smoke-induced pulmonary arterial hypertension model applied in rats. According to mean pulmonary arterial pressure, pulmonary vascular resistance, right ventricular hypertrophy index, wall thickness, and wall area data grape seed procyanidin decreases the inflammation by the PPAR-γ/COX-2 pathway [152].

In monocrotaline-induced pulmonary arterial hypertension rats, mean pulmonary arterial pressure, pulmonary vessel resistance, right ventricular hypertrophy index, percentage of medial wall thickness, percentage of medial wall area, and lung weight of wet and dry tissue ratio all decreased. The expression of endothelial nitric oxide synthase in lung tissue and plasma NO levels were raised up; the Ca^2+^ level in pulmonary arterial smooth muscle cells (PASMC) was lowered; the transcription of inflammatory factors including myeloperoxidase, interleukin (IL-1, IL-6), and tumor necrosis factor-alpha (TNF-alpha) was down-regulated in lung tissue; the nuclear factor-B pathway was also inhibited [153].

GSP improves locomotor recovery, decreases neuronal apoptosis, increases neuronal preservation, and manages microglial polarization in rats with spinal cord injuries (T9 vertebral laminectomy). Microglial polarization may be regulated by the TLR4-mediated NF-B and PI3K/AKT signaling pathways. These in vivo investigations are based on Locomotor Recovery Assessment; Terminal Deoxynucleotidyl Transferase dUTP Nick-End Labeling (TUNEL) Assay; Annexin V-FITC/PI Assays; NO assay and immunofluorescence staining: NeuN, GFAP, CD86, CD206, p-NF-κB-p65, p-AKT [154].

Neuroprotective effect of red grape (*Vitis vinifera*) seed and skin extract (GSSE) was determined in a mice model of Parkinson’s disease induced by neurotoxin 6-hydroxydopamine (6-OHDA), which causes oxidative damage and mimics the degeneration of dopaminergic neurons observed in Parkinson’s disease (PD). It was found that GSSE was effective in protecting dopamine neurons from 6-OHDA toxicity by reducing apoptosis, the level of reactive oxygen species (ROS), and inflammation; reducing the cleaved caspase-3 activity that helps inhibit 6-OHDA-induced mDA neuron death in a cellular model of PD; decreases ROS production induced by 6- OHDA in ESC-derived DA neurons; decreases phospho-NF-kB p65 activation induced by 6-OHDA in dopaminergic neurons; rescues motor deficits induced by 6- OHDA; prevents the loss of midbrain dopaminergic neurons (mDA) in a 6-OHDA mouse model of PD; prevents the loss of SOD1 level induced by 6-OHDA lesion. The biomarkers were for immunostaining: MAP2, AB5622, r tyrosine hydroxylase, caspase-3, and phosphorylated NF-kB p65 [155].

## 6. Clinical Studies of Grape Seed Extracts

Recently, there has been a significant increase in clinical trials, which further confirm the efficacy of grape seed extract in the treatment of major diseases such as various cancers, diabetes, hypertension, neurodegenerative disorders (e.g., Parkinson’s disease), etc (Table 16).

Resveratrol (RSV), a naturally existing polyphenol, has been shown to have significant antioxidant, anti-inflammatory, and anticancer properties. Regarding inflammation in a mouse model of collagen-induced arthritis, RSV was newly identified as a new treatment drug for suppressing said condition. Nonetheless, the medical advantages of RSV in the therapy of rheumatoid arthritis (RA) have not been established. The purpose of this randomized controlled clinical trial is to offer insight into the therapeutic advantages of RSV in the treatment of RA in patients at various stages of disease activity. In this randomized controlled clinical trial, 100 RA patients (68 females and 32 males) were randomly assigned to one of two groups of 50 patients each: an RSV-treated group that received a daily RSV capsule of 1 g in addition to routine care for 3 months, and a control group that received only basic care. Both groups’ clinical and biochemical markers of RA were evaluated. Clinical markers (such as the 28-joint count for swelling and tenderness) and disease activity score assessment concerning 28 joints were reported to be considerably lower in the RSV-treated group. Furthermore, serum levels of certain biochemical markers like C-reactive protein, erythrocyte sedimentation rate, undercarboxylated osteocalcin, matrix metalloproteinase-3, tumor necrosis factor-alpha, and interleukin-6 levels were considerably lower in RSV-treated individuals. The present study proposes that RSV be used as an adjuvant to standard antirheumatic medications [156].

In a randomized controlled clinical trial, 1 g RSV during a 3-month-long treatment significantly (*p* < 0.01) decreased the C-reactive protein (CRP) level in 100 rheumatoid arthritis (RA) patients, and the disease activity score assessing 28 joints erythrocyte sedimentation rate (DAS28-ESR), the erythrocyte sedimentation rate (ESR), the levels of interleukin-6 (IL-6), matrix metalloproteinase (MMP-3), rheumatoid factor (RF), tumor necrosis factor-alpha (TNF-α), undercarboxylated osteocalcin score (ucOC) were all highly significantly ameliorated (*p* < 0.001) [156].

In a clinical trial meta-analysis of grape seed extract (GSE) on diabetes and blood lipid levels, data were pooled using a random-effects model and weighted mean difference (WMD) was considered as the overall effect size. Fifty trials were included in the meta-analysis. Pooling effect sizes from studies demonstrated a significant decrease in fasting plasma glucose (FPG) (WMD): −2.01; 95% confidence interval (CI): −3.14, −0.86), total cholesterol (TC; WMD: −6.03; 95% CI: −9.71, −2.35), low-density lipoprotein (LDL) cholesterol (WMD: −4.97; 95% CI: −8.37, −1.57), triglycerides (WMD: −6.55; 95% CI: −9.28, −3.83), and C-reactive protein (CRP) concentrations (WMD: −0.81; 95% CI: −1.25, −0.38) following GSE therapy.

The grape seed had no implications on HbA1c levels, HDL cholesterol levels, or anthropometric parameters. The meta-analysis showed that consuming GSE lowered FPG, TC, LDL cholesterol, triglycerides, and CRP levels considerably [157].

The effect of grape seed extract ointment on wound healing was also investigated in cases of Cesarean section. A total of 129 women participated in this double-blind, randomized, controlled clinical trial. Participants were chosen through the convenience sampling method and were randomly assigned into three groups: 2.5% grape seed extract ointment, 5% grape seed extract ointment, and petrolatum. The REEDA scale was used to examine CS wound healing indices beforehand, 6 and 14 days after the treatment (redness, edema, ecchymosis, discharge, and approximation). 

The extract increased the synthesis of vessel enclosure growth factor (VEGF) along the wound’s edge. Furthermore, GSE demonstrated anti-inflammatory action via cytokines (TNF, IL-1, IL-6, IL-14), as well as antibacterial and antioxidant activities. On days 6 and 14 following intervention, the mean scores were 2.02, 0.52 and 0.98, 0.61 in the 5% ointment group, 2.83, 0.54 and 1.58, 0.67 in the 2.5% ointment group, and 2.91, 0.51 and 1.55, 0.74 in the petrolatum group. Whilst the 5% ointment group’s mean score was significantly dissimilar from the 2.5% ointment and petrolatum groups (*p* < 0.001), the 2.5% ointment group’s mean score was not statistically different from the petrolatum group on days 6 and 14 following intervention (*p* = 0.38 and *p* = 0.79, respectively) [158].

GSE can be successfully applied for cancer prevention, e.g., in preventing lung cancer. A modified phase I, open-label, dose-escalation clinical study was conducted to evaluate the safety, tolerability, MTD, and potential chemopreventive effects of leucoselect phytosome (LP), a standardized GSE complexed with soy phospholipids to enhance the bioavailability, in heavy active and former smokers. Bronchoscopies with bronchoalveolar lavage and bronchial biopsies were performed before and after 3 months of LP treatment. Hematoxylin and eosin stain for histopathology grading and IHC examination for Ki-67 proliferative labeling index (Ki-67 LI) were carried out on serially matched bronchial biopsy samples from each subject to determine responses to treatment. Such a treatment regimen significantly decreased bronchial Ki-67 LI by an average of 55% (*p* = 0.041), with concomitant decreases in serum miR-19a, -19b, and -106b, which were onco-miRNAs (oncomIRs) previously reported to be downregulated by GSE, including LP, in preclinical studies. In spite of not reaching the original enrollment goal of 20, our findings nonetheless support the continued clinical translation of GSE as an antineoplastic and chemopreventive agent against lung cancer.

It has been reported that GSE acts on downregulating well-known oncomIRs namely miR-19a, -19b, and -106b. Decreases in miR-19a and -19b upregulated insulin-like growth factor II receptor (IGF2R), PTEN mRNA expressions, and their respective protein products. Furthermore, GSE increased PTEN activity and decreased phosphorylation of AKT–a key procarcinogenic driver in lung cancer. Both PTEN and IGF2R are tumor suppressors and predicted targets of miR-19a and -19b. Downregulation of miR-106b resulting in upregulation of its downstream target, the tumor suppressor cyclin-dependent kinase inhibitor 1A (CDKN1A) mRNA and protein (p21) levels, further contributed to the antineoplastic effects of GSE [159].

In the continuation of the study, it was concluded that one month of leucoselect phytosome treatment significantly increased eicosapentaenoic acid (EPA) and docosahexaenoic acid (DHA), the omega-3 polyunsaturated fatty acids (n-3 PUFA) with well-established anticancer properties. Leucoselect phytosome also significantly increased unsaturated phosphatidylcholines (PC), likely from soy phospholipids in the phytosome and functioning as transporters for these PUFAs. Furthermore, 3-month leucoselect phytosome treatment significantly increased serum prostaglandin (PG) E3 (PGE3), a metabolite of EPA with anti-inflammatory and antineoplastic properties. Such increases in PGE3 correlated with reductions of bronchial Ki-67 LI (r = −0.9; *p* = 0.0374). Moreover, posttreatment plasma samples from trial participants significantly inhibited the proliferation of human lung cancer cell lines A549 (adenocarcinoma), H520 (squamous cell carcinoma), DMS114 (small cell carcinoma), and 1198 (preneoplastic cell line) [160].

According to another randomized, double-blinded, placebo-controlled clinical trial, GSE reduces significantly the risk of cardiovascular diseases in obese patients. In this trial the effects of GSE supplementation along with a restricted-calorie diet (RCD), on changes in blood lipid profile, visceral adiposity index (VAI), and atherogenic index of plasma (AIP) are investigated. Forty obese or overweight individuals (25  ≤  body mass index  <  40 kg/m^2^) were randomly assigned to receive GSE (300 mg/day) or placebo, plus RCD, for 12 weeks.

Levels of high-density lipoprotein cholesterol (HDL-C) and HDL-C/low-density lipoprotein cholesterol (LDL-C) significantly increased in the GSE group as compared with the placebo group at week 12 (*p* = 0.03 and 0.008, respectively, The investigation adjusted for age, sex, energy and saturated fatty acid intake). VAI, AIP, total cholesterol and triglyceride significantly decreased in the GSE group compared with the baseline (*p* = 0.04, 0.02, 0.01, and 0.02, respectively).

TNF and interleukin-6 may contribute to increased TG and VLDL levels and also decreased HDL-C levels. LDL-C reduction following GSE intake compared with the placebo did not remain significant, after adjusting for inflammatory markers. So it can be assumed that LDL-C more than HDL-C might be affected by inflammatory markers. GSE protects against atherosclerosis also [161].

GSE protects against insulin resistance in metabolic syndrome through the increasing of glucose transport. Participants were divided into grape seed extract (GSE) and placebo groups (*n* = 24 each) and received 100 mg/day of GSE or placebo and were placed on a weight loss diet for 8 weeks. The MD (mean difference ± SEM) of HOMA-IR between the GSE group (−1.46 ± 0.45) and the placebo group (−0.48 ± 0.47), (*p* = 0.020), and the MD of insulin between the GSE group (−7.05 ± 2.11) and the placebo group (−1.71 ± 2.12), (*p* = 0.024), were significant [162].

According to another randomized double-blind placebo-controlled clinical trial, red grape seed extract (RGSE) reduces serum paraoxonase activity and hyperlipidaemia. For 8 weeks, 70 MMH patients were given a placebo or the therapy (200 mg/day of RGSE). In the instances, there were significant elevations in the blood concentrations of apo-AI (*p* = 0.001), HDL-C (*p* = 0.001), and PON activity (*p* = 0.001) as well as significant reductions in the concentrations of TC (*p* = 0.015), TG (*p* = 0.011), and LDL-C (*p* = 0.014). The activity of PON was shown to be significantly correlated with apo-AI (r = 0.270; *p* < 0.01) and HDL-C (r = 0.45; *p* < 0.001). Significant differences between the RGSE and control groups (before and after treatment) for TC (*p* = 0.001), TG (*p* = 0.001), PON (*p* = 0.03), apo-AI (*p* = 0.001) and LDL-C (*p* = 0.002) were observed.

RGSE had the ability to significantly elevate HDL-C apo-AI levels while lowering TC, TG, and LDL-C levels. RGSE had the capacity to significantly increase the concentration of HDL-C apo-AI and lead to decreased TC, TG, and LDL-C levels. Following two months of RGSE treatment, a significant correlation between the changes in HDL-C and apo-AI values was also noted [163].

## 7. Grapeseed Oil and Polyphenols

In the grocery business, grapeseed oil is utilized for frying, salad dressings, vinegar marinades, hot oil frying, flavoring oils, and cereal frying. It is used in the cosmetics sector to make body lotions, hand creams, lip balms, body oils, sun lotions, and hair care products. Although France, Spain, and Argentina are all significant producers, Italy is where the majority of grapeseed oil gets produced.

### 7.1. The Composition of Grape Seeds

Grape seeds contain 13–19% oil, which is rich in essential fatty acids, about 11% protein, 60–70% digestible fiber, and non-phenolic antioxidants such as tocopherols and beta-carotene. In several grape varieties, 60–70% of polyunsaturated fatty acids are present in grapeseed oil. Tocopherols and β-carotene, which are non-phenolic antioxidants, are concentrated in grapeseed oil (mainly α-tocopherol and tocotrienol). In addition, phytosterols, which have been shown to have a strong antiatherosclerotic effect (β-sitosterol, stigmasterol, campesterol, and sitostanol), are present in significant amounts in grapeseed oil.

### 7.2. Location of Polyphenols in Grape Seed Cells

Polyphenols in grapes are found in the grape seed oil droplets. The oil is located within the cell walls in the form of discrete oil droplets, 0.6–2.0 μm in size. The oil droplets, which are composed of triglycerides, are covered with a phospholipid layer that binds to proteins. The stability of the droplets is ensured by the repulsive effect of proteins and the negative surface charge.

The methods for the extraction of polyphenols from grape seeds are summarized in Appendix A.

## 8. Conclusions and Future Perspectives

Grape polyphenols exert cardioprotective, anti-cancer, anti-diabetic, anti-obesity, anti-osteoarthritis, anti-neurodegenerative and anti-microbial effects both through direct anti-oxidant properties and antioxidant enzyme stimulating effects, and via modulating other signal transducers, for example inducing SIRT-1 gene, and inhibiting NFkappaB and mTOR gene expression, among other inflammatory genes (COX-2, MMPs). Thus, several patented products with high grape polyphenol content for therapeutic application and disease prevention too were developed. The present review article may contribute to further studies, therapeutic approaches and even the development of new compounds and products, too.

The physiologically active molecules of the polyphenol fraction can be selectively delivered to diseased cells by binding to dendrimers. Dendrimer binding can increase their chemiluminescence and antioxidant properties. Thus further in vivo and clinical studies are warranted to elucidate their beneficial effects in combination with active pharmaceutical ingredients or food supplements.

The number of studies on the binding of polyphenols to drug carriers, in particular, the formation of nanoscale conjugates, has increased significantly in the last decade. Effective use of polyphenols requires their substitution at higher concentrations than usual. As a major component of grape seed extract, resveratrol plays a central role in the synthesis of anti-cancer polyphenol-containing nanobioconjugates (e.g., dendrimers, polymer nanoparticles, liposomes, nanotubes, micelles, etc.) [164]. Dendrimers, as nanocarriers, can play a prominent role in this, because their multifunctional surface area allows them to contain a high local concentration of an active ingredient in a small volume, which makes their application more efficient [165].

Polyphenol dendrimers are used to enhance chemiluminescence [164,166]. Dendrimers made from gallic acid produce singlet oxygen in the presence of hydrogen peroxide. They also have chemiluminescent properties, allowing the presence of singlet oxygen to be detected in chemical systems [166]. Second-generation polyphenol dendrimers were synthesized with various core molecules and chemiluminescence was measured upon reaction with H_2_O_2_ at basic pH. High chemiluminescence was measured for all types of polyphenol dendrimers, which was 120 times higher than that of gallic acid. The intensity of chemiluminescence is strongly dependent on the distance of each branch in the structure of the polyphenol dendrimers [167]. Stilbene dendrimers have also been prepared, which also have increased photochemical activity [167,168].

The antioxidant activity of polyphenols from green tea could be significantly enhanced by enzymatic polymerization (polycatechin) or coupling to polyamino-amide dendrimers (PAMAM-catechins) [169]. First, second, and third generation dendrimers containing two, four, and eight tannic acid groups, respectively, were synthesized. The antioxidant property of the dimer is more than four times that of the monomolecular tannic acid, the tetramer more than twice that of the monomolecular tannic acid, and the octamer one and a half times [170]. The antioxidant activity of the vitamin E analog (tonox) bound to gold nanoparticles is increased eightfold compared to the free molecule [171]. Naturally occurring polyphenols bind strongly to both proteins and cell membranes. Taking advantage of this, dendrimers with a catechol-modified surface can deliver a wide variety of bioactive proteins and polypeptides into cells. Recent experimental results demonstrate that catechol dendrimers can also kill tumor cells in vivo, e.g., by transporting the enzyme alpha-chymotrypsin into the tumor cell-matrix [36].

Because of their general anti-inflammatory and free radical scavenging properties, polyphenols are likely to be widely used as medicines [172]. The studies presented demonstrate that their therapeutic and preventive role can be significant against the most common pathologies (lung cancer, atherosclerosis, hypertension, diabetes, microbial infections, etc.) [173]. The efficacy and/or targeted therapeutic use of polyphenols can also be achieved by using various nanocarriers such as dendrimers [174]. Active substances (e.g., from grape seed extract), which can be produced cheaply and in an environmentally friendly way on an industrial scale [175], could significantly replace synthetic drugs, which are not only expensive to produce but also are more toxic for the living organisms than polyphenols in general [165]. Combined with the latest therapeutic technologies (e.g., gene therapy), polyphenols could well complement and make the medicine of the future more effective [165,176].

## Figures and Tables

**Figure 1 ijms-23-11165-f001:**
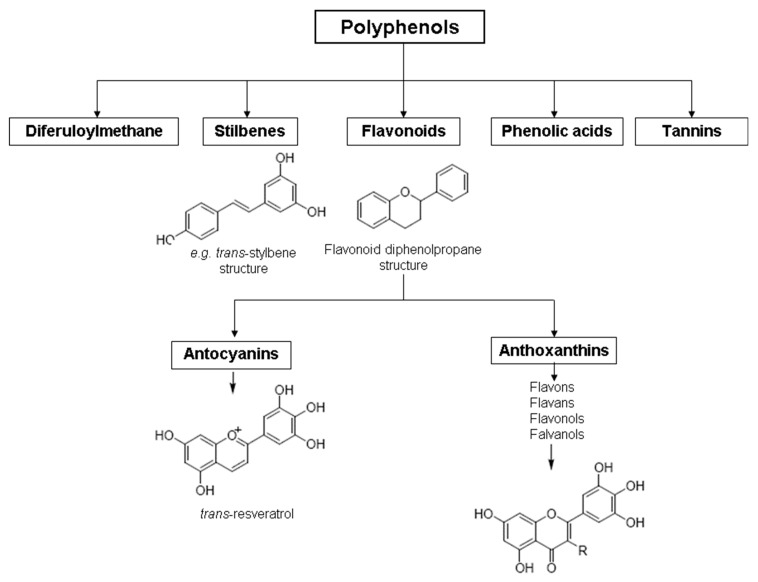
General structure of polyphenols.

**Table 1 ijms-23-11165-t001:** Main groups of polyphenols.

	Compound Group	General Structural Formula	Function	Representatives
Flavonoids	Anthocyanidins	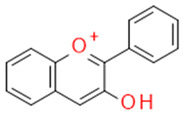	Plant dyes	Cyanidine
	Flavonols	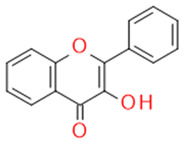	Inhibitors of drug-metabolizing enzymes	Quercetin
	Flavanols	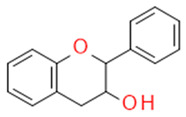	The building blocks of proanthocyanides	Catechin, epicatechin
	Isoflavonoids	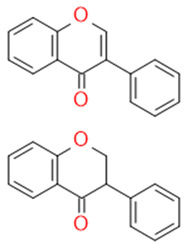	Immune booster, estrogen stimulator	Isoflavone, genistein
	Flavons	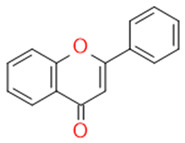	Stimulates the function of cytochrome p450	Apigenin
	Flavonones	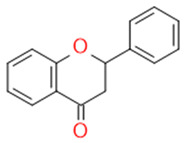	Antidiabetics	Hesperetin, Naringenin, Eriodictyol
Stilbenoid	Stilbene	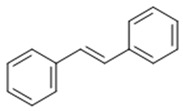	Antioxidant	Resveratrol

**Table 2 ijms-23-11165-t002:** Most important physiologically active compounds of the polyphenol fraction isolated from grape marc (grape seeds and grape skins).

Source	Compound Name	Classification	Structural Formula	Function
Grape seed and skin	Cyanidin	Anthocyanidin	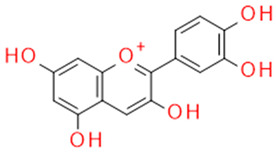	Oxygen radical sequestration
	Catechin/Epicatechin	Catechins flavan-3-ol	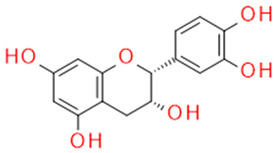	Anticancer Antisclerotic Antidiabetic Free radical sequestration
	Quercetin	Flavonol	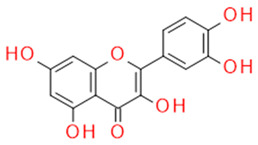	Anti-inflammatory Antiallergic Anticancer Antioxidant
Whole grapes	Resveratrol	Fitoalexin Stilbene	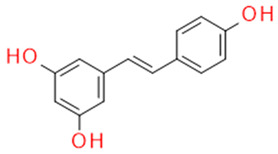	Antioxidant Antimicrobial Anticancer Anti-inflammatory Blood glucose lowering
	Rutin	Quercetin-3-rutinozide, flavonoid	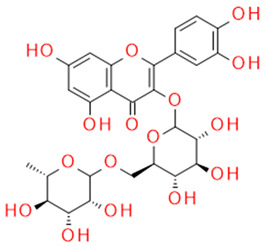	Anti-inflammatory Vasoprotective Blood clotting inhibitor Antidiabetic

**Table 3 ijms-23-11165-t003:** Physical properties of the main monomeric components of polyphenols.

Physical Properties	Catechin	EC	EGC
Molecular weight (M_r_)	293	294	445
Melting point, °C	174	236	236
Optical rotation, degree	0°	58.3°	188°
A_max_	264–280 nm		

**Table 4 ijms-23-11165-t004:** Methods used for the determination of antioxidant content.

Title	Method	Materials Needed	Literature
Antioxidant activity determination	FRAP method	FeCl_3_, triazine	[23]
András Boór total antioxidant content		2,4,6-Tris(2-pyridyl)-s-triazine	[23,27]
Determination of total polyphenol content		Folin Ciocalteu Reagent, Gallic acid, Na_2_CO_3_, Methanol	[23]
Free radical scavenging activity (antiradical activity)		1,1-Diphenyl-2-picrylhydrazine	[28]
Determination of anthocyanin content	Dilution at 550 nm with 96% ethanol containing 2% HCL at 2% *v*/*v*, followed by spectrophotometry		[24]
Determination of leucoanthocyanins	spectrophotometrically after heating with a 40:60 mixture of hydrochloric acid and butanol containing ferrous sulphate		[24]
Determination of catechin content	reacted with sulphuric acid vanillin in an alcohol-diluted solution at 500 nm by spectrophotometry	Vanillin	[25]
Resveratrol content determination	directly to HPLC		[26]

**Table 5 ijms-23-11165-t005:** Antioxidant and free radical scavenging properties of polyphenols isolated from grape marc (grape seeds and grape skins).

Polyphenol Name	Molecular Mechanism of the Protective Effect	Cell Culture	Level	Ref.
Epigallocatechin, EGCG ^1^, ECG ^2^	Lipoxygenase and cyclooxygenase inhibition	Human colon mucosa and tumor tissue	In vitro	[37]
EGCG, ECG	ARE ^3^-mediated gene expression through activation of MAPK ^4^ proteins (ERK, JNK, P38)	Hep G2 ARE in C8 cells	In vitro	[38]
Catechin, Proanthocyanidin B4	Increases CAT ^5^, GST ^6^ and SOD ^7^ activity, increases intracellular GSH ^8^ levels	Heart H9C2 cells	In vitro	[39]
EGCG, Quercetin, ECG	Inhibition of mitochondrial proton F0F1-ATPase/ATP synthase	Rat brain F_0_F_1_ ATPase	In vitro	[40]
(-)-epicatechin, procyanidin, EGCG, ECG	The recombinant human platelet Inhibition of 12-lipoxygenase and 15-lipoxygenase	J774A-1 cells	In vitro	[41]
Resveratrol	Inhibition of O-acyltransferase and sulfotransferase activity Prevention of oxidative DNA damage	Ovine ovarian tissue	In vitro	[42]
	Inhibition of H_2_O_2_ production and PMO activity Increasing GSH levels and SOD activity Reducing PMO and oxidized GR levels	Mouse skin	Ex vivo	[43]
Quercetin	Inhibits LDH cleavage Increases the activity of SOD, CAT, GSH, GPx ^9^ and GR ^10^	HepG2 cells	In vitro	[44]
	MDA and lipoperoxidation coupling Increase in Cu/Zn SOD and GPx mRNA levels	Rooster semen	In vitro	[45]
	Increasing the expression and activity of NQO1 ^11^	MCF 7 in human breast cancer cells	In vitro	[46]
	γ-GCS ^12^ level increase	Central neuron cells	In vitro	[47]
	Increasing ARE binding activity and transcriptional activity regulated by NRF2 ^13^ Activation and stabilization of NRF2 Keap 1 ^14^ reduces protein levels	Human B lymphoma cells	In vitro	[48]
	Reduction of PhIP-DNA adduct formation catalysed by O-acyl transferase and sulfotransferase	Primary culture of human mammary epithelial and adipose cells	In vitro	[49]
	Inhibits the expression and activity of CYP1A1/1A2 ^15^	In microsomes and intact Hep G2 cells	In vitro	[50]
	Inhibition of mitochondrial proton F_0_F_1_-ATPase/ATP synthase	Caco-2 cell line	In vitro	[51]

^1^ Epigallocatechin gallate; ^2^ Epicatechin gallate; ^3^ Antioxidant Response Elements (ARE); ^4^ Mitogen activated protein kinase; ^5^ catalase; ^6^ Glutathione S-transferase; ^7^ Superoxide dismutase; ^8^ Glutathione; ^9^ Glutathione peroxidase; ^10^ Glutathione reductase; ^11^ NADPH quinone oxidoreductase 1; ^12^ γ-glutamyl-cysteine synthetase; ^13^ NRF2 erythroid nuclear factor 2; ^14^ NRF2-Kelch-like ECH-associated protein 1; ^15^ Cytochrome P450-dependent monooxygenase 1A1 and 1A2.

**Table 6 ijms-23-11165-t006:** Anti-atherosclerotic and cardioprotective effects of polyphenols isolated from grape marc (grape seeds and grape skins).

Polyphenol Name	Molecular Mechanism of the Protective Effect	Cell Culture	Level	Ref.
Resveratrol	Inhibition of MMP-9 ^1^ expression and activity	Cisplatin-resistant human OSCC cell line	In vitro	[52]
	Promotion of myocardial vessel formation by induction of VEGF ^2^, Trx-1 ^3^ and HO-1 ^4^	H9C2 cells	In vitro	[53]
	Inhibition of the expression and binding activity of MCP-1 ^5^ and CCR2 ^6^ receptors	Endometriotic stomal cells	In vitro	[54]
	Increase NO and NOS levels Increasing intracellular cGMP levels and reducing ANP ^7^ and BNP ^8^ levels	U2OS cells	In vitro	[55]
	Reduces monocyte cell adhesion to stimulated endothelium Reduces VCAM-1 ^9^ mRNA and protein formation	Human vascular endothelial cells	In vitro	[56]
EC	7β-OH inhibition of cholesterol formation	Smooth muscle cells	In vitro	[57]
Quercetin	Increase serum LDL-bound PON-1 ^10^ levels	HuH7 in human liver cell line	In vitro	[58]
	Induction of IFN-γ ^11^ gene expression Inhibition of IL-4 ^12^ gene expression	Peripheral blood in Human Peripheral-blood CD4+ T cells	In vitro	[59]
	Increase in intracellular GSH levels and activation of the γ-GCS ^13^ heavy subunit (GCS(h)) promoter	Central neuron cell line	In vitro	[47]
Genistein Daidzein	They are incorporated into LDL, increasing its resistance to oxidation and its effectiveness in inhibiting cell proliferation	Human colon cancer cell line	Ex vivo, in vitro	[60]
EGCG, EGC	Inhibition of rat VSMC ^14^ precipitation on collagen and laminin Interference with VSMC integrin β1 receptor and ECM protein binding	Rat VSMC	In vitro	[61]
Procyanidins	Reducing the leukotriene-to-prostacyclin ratio in blood plasma	Human aortic endothelial cells	In vitro	[62]
Proanthocyanidin	Inhibition of CD36 mRNA expression	THP-1 cells	In vitro	[63]

^1^ Matrix metalloproteinase 2; ^2^ Vascular endothelial growth factor; ^3^ Thioredoxin-1; ^4^ Hem oxygenase-1; ^5^ Monocyte chemotactic protein-1; ^6^ Chemokine receptor-2; ^7^ Pitvar natriuretic peptide; ^8^ Brain natriuretic peptide; ^9^ Vascular cell adhesion molecule-1; ^10^ Paraoxonase-1; ^11^ γ-interferon; ^12^ interleukin-4; ^13^ γ-glutamylcysteine synthetase; ^14^ Vascular smooth muscle cell.

**Table 7 ijms-23-11165-t007:** The protective effects of polyphenols isolated from grape marc (grape seeds and skins) on the nervous system.

Polyphenol Name	Molecular Mechanism of the Protective Effect	Cell Culture	Level	Ref
Resveratrol	Stimulates AMP kinase activity	Neuro2a in cells and primary neurons; MC3T3-E1 cells and primary osteoblasts	In vitro	[64]
	Activation of phosphorylation of PKC Transcitrin selection to prevent Aβ1 aggregation ^1^	Rat hippocampal cell culture; endothelial cell culture	In vitro	[65]
	Protection of dopaminergic neurons Activation of the sirtuin family of NAD-dependent histone deacetylases	Organotypic mid-brain slice culture; human umbilical vein endothelial cells	In vitro	[66]
EGCG, ECG, Myricetin	Inhibition of IL-6, IL-8, VEGF and PGE2 ^2^ production Attenuation of COX-2 expression and NF-κB ^3^ activation Induction of MAPK phosphatase 1 expression Inhibition of phosphorylation of MAPK (p38 and JNK ^4^)	Human astrocytoma U373MG cell culture	In vitro	[67]
	Attenuation of mitochondrial membrane potential rupture and release of CYT-C ^5^ Reducing caspase-9 and caspase-3 activity and increasing the BAX:BCL-2 ratio	Rat PC12 cells; HeLa cell line	In vitro	[68]
Epicatechin	Protects neurons from programmed cell death induced by oxLDL ^6^ by inhibiting the activation of JNK, c-JUN and caspase-3	Primer neuron cell culture	In vitro	[69]

^1^ Amyloid beta aggregation; ^2^ Prostaglandin E2; ^3^ Nuclear Factor-κB; ^4^ C-JUN terminal kinase; ^5^ Cytochrome c; ^6^ Oxidized LDL.

**Table 8 ijms-23-11165-t008:** Anti-inflammatory effects of the polyphenol content of grape marc (grape seeds and skins).

Polyphenol Name	Molecular Mechanism of the Protective Effect	Cell Culture	Level	Ref.
Procyanidins	Inhibition of IL-1β transcription and secretion	ARPE-19 cells	In vitro	[72]
EGCG, ECG	Inducing programmed cell death by activating caspases 3, 8 and 9	Caco-2 cells	In vitro	[73]
	Inhibition of CD11b expression Inhibition of peripheral CD8+ T-cell migration and proliferation	HepG2 cells	In vitro	[74]
Resveratrol	Inhibition of caspase-3 stimulation and IL-1β -induced cleavage of PARP	SH-SY5Y cells	In vitro	[75]
	Inhibition of iNOS mRNA and protein expression by inhibiting NF-κB activation Inhibition of NO production	murine microglial cell line N9	In vitro	[76]
	Activation of MAP kinase phosphatase	Prostate cells	In vitro	[77]
Quercetin	Blocking the expression of ICAM-1 ^1^, VCAM-1, and E-selectin Inhibition of PG synthesis and IL-6, 8 productions	HUVECs	In vitro	[78]
	Inhibition of THP-1 adhesion and VCAM-1 expression activation	ARPE-19 cells	In vitro	[79]
	Inhibition of NO production and inhibition of iNOS ^2^ protein expression	hep g2 cells	In vitro	[80]
Anthocyanins	Localization in endothelial cells Reduction of IL-8, MCP-1 and ICAM-1 activation	Caco-2 cells	In vitro	[81]

^1^ Intracellular adhesion molecule-1; ^2^ Inducible nitric oxide synthase.

**Table 9 ijms-23-11165-t009:** Mutation-reducing/anti-cancer effects of the polyphenol content of grape marc (grape seeds and skins).

Polyphenol Name	Molecular Mechanism of the Protective Effect	Cell Culture	Level	Ref.
Resveratrol	Inhibition of cell proliferation and reduction of telomerase activity	Human cancer cell line HCT116	In vitro	[86]
	Stimulation of the P53-dependent pathway of programmed cell death	Human lung adenocarcinoma cells A549	In vitro	[87]
	Inhibition of cell proliferation by interaction with the ERα ^1^-related PI3K pathway	Estrogen-sensitive MC3T3-E1 precursor cells	In vitro	[88]
	Inhibition of COX-2 expression through inhibition of MAPKs and AP-1 activation	RAW 264.7 macrophages	In vitro	[89]
	Reduction of expression of COX-1, COX-2, c-MYC, c-FOS, c-JUN, TGF-β 1 ^2^ and TNF-α	Mucosal cell line	In vitro	[90]
	Inhibits oncogenic diseases through inhibition of protein kinase CKII activity	Human breast cancer mcf-7 cells	In vitro	[91]
	Inhibition of PKCα and PKCβI Ca^2+^-dependent activity	Smoth muscle cells	In vitro	[92]
	Prevents the formation of NB ^3^-DNS and NB-Hb ^4^ adducts	Hemoglobin of mice	In vivo	[93]
Quercetin	Blocking EGFR tyrosine kinase activity	Xenografted NSCLC cells EGFR C797S mutation	In vitro	[94]
Quercetin, Myricetin	Inhibition of human CYP1A1 activity Inhibition of DE2 ^5^ formation and B[a]P activation	O-deethylation of 7-ethoxyresorufin human lymphoblastoid TK6 cells	In vitro	[49]
Quercetin	Interaction with glycoprotein P and regulation of BCRP/ABCG2 ^6^ activity	In two different cell lines expressing BCRP	In vitro	[95]
EGCG	Telomerase inhibition	In human cancer cells HeLa	In vitro	[96]

^1^ Estrogen receptor α; ^2^ Transformational growth factor 1; ^3^ Nitrobenzene; ^4^ Hemoglobin; ^5^ Diol epoxide 2; ^6^ ATP-binding cassette transporter for breast cancer resistance protein.

**Table 10 ijms-23-11165-t010:** Effect of the polyphenol content of grape marc (grape seeds and skins) on signal transduction.

Polyphenol Name	Molecular Mechanism of the Protective Effect	Cell Culture	Level	Ref.
Proanthocyanidins	Accelerate programmed cell death by altering the cdki-cdk-cyclin cascade and reducing mitochondrial membrane potential through activation of cascade 3	Human epidermoid carcinoma A431 cells	In vitro	[97]
Quercetin	Inhibition of phosphorylation of JNK and P38 MRK by ROS ^1^-mediated signaling	Murine macrophage cell line RAW 264.7	In vitro	[98]
	Actin/PKB and ERK1/2 signaling cascade to affect neuronal functionality	P19 neuronal cells	In vitro	[99]
Resveratrol	Inhibits monocyte NO, MAPK and PI3K-dependent CCR2 binding	Rat fibroblast-like synoviocyte RSC-364 cell line	In vitro	[100]
	Inhibit cardiac fibroblast division via NO-cGMP signaling	Rat heart in fibroblast culture	In vitro	[101]
	Activates phase II genes through regulation of ARE/EpRE activation Modifies the performance of KeapI by binding NRF2	Lung cancer cells	In vitro	[102]

^1^ Reactive oxygen species.

**Table 11 ijms-23-11165-t011:** Effect of the polyphenol content of grape marc (grape seeds and grape skins) on endothelial cells and blood vessel walls.

Polyphenol Name	Molecular Mechanism of the Protective Effect	Cell Culture	Level	Ref.
EGCG, Quercetin	Inhibition of programmed cell death through regulation of BCL-2 and BAX Inducing nuclear transactivation of P53 Reducing the activity of caspase 3 Blockade of JNK and P38 MARK-related singletons	3T3-L1 preadipocytes	In vitro	[103]
Cy3G ^1^	Increases eNOS expression and activity NO production triggering Regulation of phosphorylation of eNOS and AKT increase cGMP production	Endothelial cell line	In vitro	[104]
EGCG	Endothelium-dependent vasodilator effect Activates phosphatidylinositol 3-kinase, AKT, and eNOS.	HUVEC	In vitro	[105]
	Increases the activity of eNOS Induces continuous activation of AKT, ERK1/2, and eNOS Phosphorylation of Ser1179	Calf aortic endothelial cells	In vitro	[106]
Catechins	Chicken CAM ^2^ angiogenin-like protein reduces angiogen-induced vascularization	In chicken cells	In vitro	[107]
Proanthocya-nidin	Reducing VCAM-1 expression Reduces TNFα-induced T cell binding to HUVEC	Primary HUVEC	In vitro	[108]
Procyanidine, flavan-3-ols	They inhibit the activity of ACE ^3^	Two substrates	In vitro	[13]

^1^ Cyanidin-3-glucoside; ^2^ Chicken chorioallantoic (embryonic spinal cord) membrane; ^3^ Angiotensin-converting enzyme.

**Table 12 ijms-23-11165-t012:** Effect of the polyphenol content of grape marc (grape seeds and skins) on diabetes.

Polyphenol Name	Molecular Mechanism of the Protective Effect	Cell Culture	Level	Ref.
EGCG, ECG	Inhibits SGLT1 and sodium-free GLUT	Polarized Caco-2 intestinal cells	In vitro	[109]
Quercetin	Reduces blood sugar levels Inhibits SVCT1 ^1^ and GLUT2	Intestinal cell model	In vitro	[110]
Tannin, anthocyanin	Inhibition of α-amylase and α-glucosidase	On 2-chloro-4-nitrophenyl-4-O-β-D-galactopyranosyl maltosyl substrate	In vitro	[111]

^1^ Na-dependent vitamin C transporter 1.

**Table 13 ijms-23-11165-t013:** Effect of the polyphenol content of grape marc (grape seeds and skins) on the cell cycle.

Polyphenol Name	Molecular Mechanism of the Protective Effect	Cell Culture	Level	Ref.
Resveratrol	Stimulates P21 expression and arrests the cell cycle in G1 phase	A375SM malignant melanoma	In vitro	[112]
	Inhibition of cyclin D1/D2-cdk6 cyclin D1/D2-cdk4 cyclin E-cdk2 complexes	MCF7 cells	In vitro	[113]
	Decreases cyclin D1/Cdk4 complex and stimulates expression of cyclin E and A	Melanoma cells	In vitro	[114]
	Decrease the hyperphosphorylated form of pRb and increase the hypophosphorylated form of pRb Decrease expression of E2F (1–5) transcription factors and their heterodimer partners DP1, DP2 Leads to cell cycle arrest in the G0/G1 phase	Embryonic rat heart cell line	In vitro	[115]
Proanthocyani dines	Inhibit expression of cyclin B1, D1, A1 and 𝛃-catenin	Human cancer cell lines	In vitro	[116]
	They stop the cell cycle in the G1-S phase	VMSC at human hepatocellular carcinoma cells	In vitro	[117]

**Table 14 ijms-23-11165-t014:** Other bioactive effects of the polyphenol content of grape marc (grape seeds and skins).

Type of Activity	Polyphenol Name	Molecular Mechanism of the Protective Effect	Cell Culture	Level	Ref.
Anti-HIV effect	Proanthocyanidins	Inhibits expression of the HIV-preventing chaperones CCR2b, CCR3, and CCR5.	Normal peripheral mononuclear cells	In vitro	[125]
Sensory effect	Proanthocyanidins, Resveratrol	Enhancing VEGF expression	Pigment cell culture; retinal ARPE-19 cells	In vitro	[124]
Liver protection	Genistein	Reduces experimental liver damage by preventing lipid peroxidation and enhancing the antioxidant system	Rat and Human hepatocyte-derived cell lines (ie HepG2 and Hep3B)	In vitro	[127]

**Table 15 ijms-23-11165-t015:** In vivo experiments for investigations of healing effects of grape seed extract and its components in different diseases.

Polyphenol Name	Molecular Mechanism of the Protective Effect	Target Organ/Disease	Type of Investigation	Biomarker	Animals	Ref.
Lipophilic Grape Seed Proanthocyanidin (LGSP)	Apoptosis via decreasing the expression of cyclin D1 and CDK 4 and increasing the expression of the tumor suppressors p21 and p27; activation of cleaved fragments of caspases 3, caspases 9, and PARP	PC3 Human Prostate Cancer Cell xenograft	xenograft model via oral gavage LGSP	Ki67 and cleaved caspase 3 immunostaining	PC3-derived mouse	[150]
Grape Seed Proanthocyanidin (GSP)	GSP induces autophagy, and inhibition of autophagy increased apoptosis in HepG2 cells; inducing the phosphorylation of mitogen-activated protein kinase (MAPK) pathway-associated proteins (p-JNK, p-ERK and p-p38 MAPK); reduces the expression of survivin	HepG2 (human liver cancer cells)-derived xenografts	xenograft model via oral gavage GSP	Ki67 immunostaining	nude mouse	[151]
Grape Seed Procyanidin	decrease the inflammation by PPAR-γ/COX-2 pathway	Pulmonary arterial hypertension model	treated with normoxia/cigarette smoke	mPAP, PVR, RVHI, WT%, and WA% was detected in the rats	Sprague Dawley rats	[152]
Grape Seed Proanthocyanidin (GSP)	endothelial nitric oxide synthase expression in lung tissue and plasma NO level were increased; Ca^2+^ level in pulmonary arterial smooth muscle cell (PASMC) was decreased; transcription of inflammatory factors such as myeloperoxidase, interleukin (IL)-1β, IL-6 and tumor necrosis factor alpha (TNF-α) was down-regulated in lung tissue; nuclear factor-κB pathway was inhibited as IκBα was less phosphorylated; TNFα-induced PASMC overproliferation could be inhibited	Pulmonary arterial hypertension model	treated with monocrotaline	Haemodynamic index, mean pulmonary arterial pressure (mPAP), cardiac output (CO), pulmonary vessel resistance (PVR), right ventricular hypertrophy index (RVHI), WT%, WA%, pulmonary blood pressure NO assay, cytosolic Ca^2+^ detection	Sprague Dawley rats	[153]
Grape Seed Proanthocyanidin (GSP)	promoted locomotor recovery, reduced neuronal apoptosis, increased neuronal preservation, and regulated microglial polarization; microglial polarization and prevents neuronal apoptosis, possibly by the TLR4-mediated NF-κB and PI3K/AKT signaling pathways	Spinal cord injury	T9 vertebral laminectomy	Locomotor Recovery Assessment; Terminal Deoxynucleotidyl Transferase dUTP Nick-End Labeling (TUNEL) Assay; Annexin V-FITC/PI Assays; NO assay, Immunofluorescence staining: NeuN, GFAP, CD86, CD206, p-NF-κB-p65, p-AKT	Sprague Dawley rats	[154]
Red grape seed and skin extract	GSSE was effective in protecting dopamine neurons from 6-OHDA toxicity by reducing apoptosis, the level of reactive oxygen species (ROS) and inflammation; reducing the cleaved caspase-3 activity that helps inhibit 6-OHDA-induced mDA neuron death in a cellular model of PD; decreases ROS production induced by 6- OHDA in ESC-derived DA neurons; decreases phospho-NF-kB p65 activation induced by 6-OHDA in dopaminergic neurons; rescues motor deficits induced by 6- OHDA; prevents the loss of midbrain dopaminergic neurons (mDA) in a 6-OHDA mouse model of PD; prevents the loss of SOD1 level induced by 6-OHDA lesion	Parkinson’s disease	neurotoxin 6-hydroxydopamine (6-OHDA), which induces oxidative damage and mimics the degeneration of dopaminergic neurons observed in Parkinson’s disease	Immunostaining: MAP2, AB5622, r tyrosine hydroxylase, caspase-3, phosphorylated NF-kB p65; ROS assay,	mice	[155]

**Table 16 ijms-23-11165-t016:** Clinical investigation of grape seed extract polyphenols as therapeutics against the most common diseases.

Polyphenol Name	Molecular Mechanism of Therapeutic Effect	Target Organ/Disease	Type of Investigation	Biomarker	Patients	Ref.
Resveratrol	STAT3/HIF-1/VEGF pathway	Rheumatoid arthritis	Randomized controlled clinical trial	CRP, DAS28-ESR, ESR, IL-6, MMP-3, RF, TNF-α, ucOC	100	[156]
Grape seed extract	Reduces FPG, TC, LDL cholesterol, and triglycerides levels;	Glycemic control	Randomized controlled clinical trial	serum TC, LDL, VLDL, HDL colesterol, triglycerides level	50	[157]
Grape seed extract	Suppress lipoxygenase pathways; increase pro-inflammatory leukotrienes	Inflammation	Randomized controlled clinical trial	CRP, pro-inflammatory leukotrienes, cytokine pattern	50	[157]
Grape seed extract	VEGF, anti-inflammatory activity through cytokines (TNF, IL-1, IL-6, IL-14), antibacterial activity, antioxidant activity	Wound healing after Cesarean section	Randomized controlled clinical trial	REEDA scale (redness, edema, ecchymosis, discharge, and approximation)	129	[158]
Grape seed procyanidin extract	inhibit the proinflammatory and procarcinogenic COX-2/PGE2 pathways; 15-lipoxygenase (15-LOX) and 15-Hydroxyeicosatetraenoic acid (15-HETE) pathways	Lung cancer	Randomized controlled clinical trial	Ki67 proliferative labeling index; serum miR-19a, -19b, and -106b	287 (146/control 141)	[159]
Grape seed procyanidin extract	COX-2/PGE2 pathways	Lung cancer	Randomized controlled clinical trial	Serum PGE3 and leukotriene B5 (LTB5)	287	[160]
Grape seed extract	Reduces TNF and IL-6 level, and TG and VLDL level decreases and HDL-C level increases. It protects against atherosclerosis	Cardiovascular prevention in obesity	Randomized, double-blinded, placebo-controlled clinical trial	visceral adiposity index (VAI), and atherogenic index of plasma (AIP); plasma LDL-C level	50 (25/25)	[161]
Grape seed extract	Increases glucose transport	insulin resistance in metabolic syndrome	Randomized controlled clinical trial	Plasma FBG, TG, HDL-C and insulin level	48 (24/24)	[162]
Red grape seed extract	Reduces TNF and IL-6 level, TG and VLDL level decreases, and HDL-C level increases.	hyperlipidaemia	Randomized controlled clinical trial	apolipoprotein AI and paraoxonase activity	70	[163]

## Data Availability

Not applicable.

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
