# Peer review of "The Utilization of Physiologically Active Molecular Components of Grape Seeds and Grape Marc"

_ijms, 2022, doi:10.3390/ijms231911165_

Round 1
Reviewer 1 Report
I found the review by Hegedüs et al. Interesting. The methodology with which it was drafted was mentioned. The logical approach of the manuscript is appreciable and the arguments well described and complete. The degree of detail of the information is excellent.
Author Response
Answer to Reviewer 1.
Dear Reviewer,
thank you for your favourable review. We have improved the quality of the manuscript by adding new tables and updating the bibliographical references.
Thank you for your kind help
Sincerely:
the authors
Reviewer 2 Report
The review paper is aimed about potential physiologically active components of grapes, especially seeds and marc. The authors presented quite extensive information on the issue. The division of the article is logical. However, my main criticism is a lack of novelty. The authors should highlight the novelty.
In the overall evaluation, it is necessary to point out the low topicality of the presented review. For example, if we look at the references, the use of only 25 literary sources out of 241 published in the last 5 years is clear. We can classify the remaining literary sources used as 20-30 years old, without subsequent continuous research activity. Which, of course, does not reduce the quality of older studies, but creates room for speculation in terms of topicality and novelty. It is rather unexpected to find that new information is used in the introduction, but it is absolutely absent in important/interesting sections such as e.g. “Anti-atherosclerosis and cardioprotective effects; Neuroprotective effects; Anti-inflammatory effect; Mutation reduction and anti-cancer effect; Effects on the vascular wall and choroidal cells; Effects on the cell cycle”, etc. Rather, I would expect the opposite tendency. As an example, I can use the keywords "resveratrol and cancer" on Web of Science with 1200 results in the last 2 years! The authors should highlight the novelty.
Specific comments:
1. Introduction and review methodology
This section is structurally and content sufficiently processed. It is not necessary to add new references here.
Page 1 line 47: It is not necessary to give the same reference (3) in two consecutive sentences.
Page2 line 65: Google Scholar is not database. Google scholar is web search engine which works on metadata and refers to various databases.
Page 2 line 70: In the methodology, it would be appropriate to provide absolutely accurate information. Please avoid cases of "e.g. resveratrol" but include all keywords.
2. Physical and chemical properties of polyphenols
I recommend renaming or redesigning this section. By changing the title to: “Basic physical and chemical...” you will put things in perspective. I justify my request with the missing other properties of polyphenols (especially chemical properties).
3. Polyphenol effects on the human body
In this part of the presented work, the absence of new sources is very noticeable. Specifically, section “3.1 Antioxidant and free radical scavenging activity” in Table 5, only the results of studies published 20 years or more ago are actually used. Again, I do not claim that the sources are not relevant, but it is necessary to supplement them with the latest experimental studies, of which there are an immense number both in vitro and in vivo.
Page 8 and 9 - description to table 5 - incorrect names: 1Epigallokatekin; 2Epikatekin; correct names: Epigallocatechin; Epicatechin
In sections 3.2 Anti-atherosclerosis and cardioprotective effects; 3.3 Neuroprotective effects; 3.4 Anti-inflammatory effect; 3.5 Mutation reduction and anti-cancer effect; 3.6 Influencing signal transduction; 3.7 Effects on the vascular wall and choroidal cells; 3.8 Effects on diabetes; 3.9 Effects on the cell cycle; and 3.10 Other impacts, there is the same problem as in the previous one (3.1). I have to use an example again, searching the keywords "resveratrol and diabetes" on Google Scholar I found 17 200 results in the last 2 years! The authors should highlight the novelty.
Furthermore, I recommend that the authors give a more appropriate title for this chapter. “Polyphenol effects on the human body” the title does not fully describe what the authors were working on. It will either be necessary to use only human studies, or to divide the section into in vivo and in vitro.
Sections 4 and 5 are sufficiently described in my opinion.
Section 8 "Summary": I don't understand why it is numbered eight? Please edit. Again, I am not completely identified with what the authors summarize in this section. The conclusion acts more like the introduction of the article. I think that the possibilities of extraction, properties such as taste, or use in the food industry, are irrelevant for this review. Authors should focus on the molecular essence and the essence of the effect of polyphenols at the molecular level, as well as their possible use against diseases, focus on novelty and possible future use and visions in molecular biology, medicine or ethnopharmacy.
Author Response
Answers to Reviewer 2.
thank you for your valuable and thorough review. The manuscript was reviewed by the authors, most of the literature was refreshed and new chapters are now inserted to the text.
Reviewer’s comment:
Specific comments:
- Introduction and review methodology
This section is structurally and content sufficiently processed. It is not necessary to add new references here.
Page 1 line 47: It is not necessary to give the same reference (3) in two consecutive sentences.
Answer: the given reference was removed.
Reviewer’s comment:
Page2 line 65: Google Scholar is not database. Google scholar is web search engine which works on metadata and refers to various databases.
Answer: This expression has exchanged:
„…Web of Science and PubMed databases were used, augmented with Google Scholar search engine.”
Reviewer’s comment:
Page 2 line 70: In the methodology, it would be appropriate to provide absolutely accurate information. Please avoid cases of "e.g. resveratrol" but include all keywords.
Answer: The given sentence has changed and compared:
„Later on, these keywords were supplemented with keywords for the most typical ingredients (e.g. “resveratrol”, “quercetin”, “tannin”, “anthocyanin”) or the most prominent physiological effects (e.g. “antioxidant”, “free radical scavenger”, “Anti-atherosclerotic”, “cardioprotective”, “nervous system”, “anti-inflammatory”, “anti-cancer”, “signal transduction”, “endothel”, “blood vessel”, “diabetes”, “cell cycle”, “bioactive”, “in vivo”, “clinical”, “preventive”, “therapeutic”) and these results were compiled.”
Reviewer’s comment:
- Physical and chemical properties of polyphenols
I recommend renaming or redesigning this section. By changing the title to: “Basic physical and chemical...” you will put things in perspective. I justify my request with the missing other properties of polyphenols (especially chemical properties).
Answer: the suggested renaming has replaced.
Reviewer’s comment:
- Polyphenol effects on the human body
In this part of the presented work, the absence of new sources is very noticeable. Specifically, section “3.1 Antioxidant and free radical scavenging activity” in Table 5, only the results of studies published 20 years or more ago are actually used. Again, I do not claim that the sources are not relevant, but it is necessary to supplement them with the latest experimental studies, of which there are an immense number both in vitro and in vivo.
Answer: the full chapter has reviewed and almost the all references has replaced to newer ones from the last 3-5 years.
Reviewer’s comment:
Page 8 and 9 - description to table 5 - incorrect names: 1Epigallokatekin; 2Epikatekin; correct names: Epigallocatechin; Epicatechin
Answer: the requested clarification has been made here and the spelling of all chemical names in the full manuscript has been checked.
Reviewer’s comment:
In sections 3.2 Anti-atherosclerosis and cardioprotective effects; 3.3 Neuroprotective effects; 3.4 Anti-inflammatory effect; 3.5 Mutation reduction and anti-cancer effect; 3.6 Influencing signal transduction; 3.7 Effects on the vascular wall and choroidal cells; 3.8 Effects on diabetes; 3.9 Effects on the cell cycle; and 3.10 Other impacts, there is the same problem as in the previous one (3.1). I have to use an example again, searching the keywords "resveratrol and diabetes" on Google Scholar I found 17 200 results in the last 2 years! The authors should highlight the novelty.
Answer: the full chapter has reviewed and almost the all references has replaced to newer ones from the last 3-5 years.
Reviewer’s comment:
Furthermore, I recommend that the authors give a more appropriate title for this chapter. “Polyphenol effects on the human body” the title does not fully describe what the authors were working on. It will either be necessary to use only human studies, or to divide the section into in vivo and in vitro.
Answer: The title of this chapter has changed:
„4. The beneficial effects of polyphenols on health and its molecular mechanisms”
Reviewer’s comment:
Section 8 "Summary": I don't understand why it is numbered eight? Please edit. Again, I am not completely identified with what the authors summarize in this section. The conclusion acts more like the introduction of the article. I think that the possibilities of extraction, properties such as taste, or use in the food industry, are irrelevant for this review. Authors should focus on the molecular essence and the essence of the effect of polyphenols at the molecular level, as well as their possible use against diseases, focus on novelty and possible future use and visions in molecular biology, medicine or ethnopharmacy.
Answer: the entire chapter has been rewritten to emphasise physiological effects, clinical applications, drug-like applications using the latest nanotechnology, and future possibilities. A renumbering was performed over the whole text.
With the best regards,
Dr Imre Hegedus,
In the name of the Authors

Reviewer 3 Report
In this manuscript, Hegedus et al summarize the different functions of the physiologically active molecular components of grape seeds and grape marc. This work is complete and exposes in a linear way the action of the different components. A graphical abstract would be welcome to close this synthesis.
Author Response
Answer to Reviewer 3.
Comments and Suggestions for Authors
In this manuscript, Hegedus et al summarize the different functions of the physiologically active molecular components of grape seeds and grape marc. This work is complete and exposes in a linear way the action of the different components. A graphical abstract would be welcome to close this synthesis.
Answer:
Dear Reviewer 3,
thank you so much for your kind review. The requested graphical abstract has been attached to the reviewed manuscript.
Best regards.
on behalf of the authors:
Imre Hegedus
Submission Date
25 July 2022
Date of this review
16 Aug 2022 10:34:03
Reviewer 4 Report
The manuscript ‘The utilization of physiologically active molecular components of grape seeds and grape marc’ needs a major revision.
Title: Remove ‘(review)’
Line 29: add the scientific name of the grape.
Line 44: add the scientific name of the grape and add the bioactive compounds name.
Separate introduction and review methodology and improve introduction.
Add conclusion and future studies.
References
Add available DOI to the reference list.
Reference 196: add English names and journal.
Author Response
Answers to Reviewer 4.
Dear Reviewer 4,
thank you for your valuable comments, the manuscript has been corrected as follows:
Reviewer’s comment:
Title: Remove ‘(review)’
Answer: word „(review)” has been removed from the title.
Reviewer’s comment:
Line 29: add the scientific name of the grape.
Answer: the text has added: „Grapes (Vitis vinifera)…”
Reviewer’s comment:
Line 44: add the scientific name of the grape and add the bioactive compounds name.
Answer: the text of the abstract has been rewritten, but the desired additions have been inserted:
„The current review concentrates on presenting and classifying polyphenol bioactive molecules (resveratrol, quercitin, catechin/epicatechin, etc.) available in high quantities in Vitis vinifera grapes or their byproducts.”
Reviewer’s comment:
Separate introduction and review methodology and improve introduction.
Answer: Introduction and Review methodology are separated into two different chapter and their text has expanded:
„1. Introduction and review methodology
Vitis vinifera Ggrapes are extremely rich in bioactive components [1]. Grape marc is a mixture of grape seeds and skins, which remain as a by-product of the wine production process, making up 20-25% of the grape's weight [2]. Grape seeds contain fats, proteins, carbohydrates, and 5-8% polyphenols. The grape seed is rich in extractable phenolic antioxidants such as phenolic acids, flavonoids, proanthocyanidins, and resveratrol, and the grape skin is abundant in anthocyanins [3]. Grape marc also contains a large amount of lipids, proteins, indigestible fibers, and minerals [1, 2].
Around 1.000 kg of grapes isare used to produce 750 literslitres of wine. By way of comparison of start- and end-product masses, this means that about 60% of the grape harvest mass will become agricultural waste [4]. As an example, in 2017, Chinese grape production was 13.083.000 tons and South African grape production was 2.032.582 tons [5]. Hence, there is a huge untapped potential in the use and extraction of active substances from grape seeds, skins, and pomace.
1.12 Polyphenols
Polyphenols are so-called secondary metabolites of plants, biologically active compounds in order to enhance plants adaptation to environmental conditions, for example balancingto balance oxidative stress [6]. Polyphenols are plants’ active substances consisting of more than one phenolic group. In food, more than 15 classes of polyphenols can be found [7]. The polyphenols are largely flavonoids, which can be further subdivided into 13 subclasses, whereith more than 8.000 components have been described. Flavonoids are the largest and most-best studied group of phenols. Their seven main subclasses are: flavones, flavonones, flavonols, isoflavones, anthocyanidins/anthocyanins, flavanols (or catechins and procyanidins), and chalcones [7]. Another group of flavonoids not included in this list are proanthocyanidins, also known as procyanodins, condensed tannins, or oligomeric procyanidins [7]. High molecular weight (from 500 D up to 20.000 D) polyphenols are plant tannins [8]. Polyphenols can generally be subdivided into hydrolyzsable tannins (tannic acid esters with glucose or other sugars) [9], phenylpropanoids (lignins, flavonoids) [10-12], and condensed tannins [13]. Polyphenol compounds, especially procyanidins, contribute to the bitter and astringent taste of juices shaping the aroma of wines [14]. The coloring agents from the grape skins are considered “generally recognized as safe” (GRAS) and are utilized as food colorants [14].
In grapes, flavonoids are mainly found in the seeds, fruit skins, and stems. Between 60 and 70 % of the total recoverable polyphenols in grapes are in the seed, which accounts for 5-8 % of the weight of the seed [15]. Hundreds of polyphenolic compounds are present in wine, which influence the taste, color, and flavor of the wine [14]. The extractable phenolic antioxidants account for 10-11% of the dry weight of the grape marc. The polyphenolic composition of marc is varietal. Red grapes are richer in proanthocyanidins, while in white grapes they are scarcely present. The composition of polyphenols depends on the grape variety, the weather, the place of cultivation, and the maturity of the grapes [16]. The largest and best- known constituents of polyphenols are flavonoids [17] (Figure 1). The vast majority of polyphenols in grape seeds are flavonoids [18]. The classification of polyphenols and the characteristic functions of each molecular class are summarized in Table 1.
The main polyphenolic constituents in grape seeds are catechins (catechin, epicatechin, procyanidin [19]). Except for epicatechin, they are found in the outer, soft layer of the grape seed. The most physiologically important compounds of polyphenols isolated from grape seeds are summarized in Table 2.
- Review methodology
Our aim was to prepare a scoping review to demonstrate that there is a significant amount of active substances in grapes, mainly in the seeds and pomace, which in many cases become waste. We also provide an overview of the wide range of physiological effects of these available active substances. Therefore, the extraction and use of these molecules as food supplements or possibly as novel pharmaceutical concepts such as dendrimer nano-bioconjugates could have a significant health-enhancing and disease-preventive effect on the population.
To access relevant articles the Web of Science and PubMed databases were used, augmented with Google Scholar search engine. The “polyphenols” and “grape” keywords were applied, and 5.981 results have been found onin Web of Science, 3.690 results on Pubmed, and more than 131.000 articles, dissertations, and scientific reports in Google Scholar.
Later on, these keywords were supplemented with keywords for the most typical ingredients (e.g. “resveratrol”, “quercetin”, “tannin”, “anthocyanin”) or the most prominent physiological effects (e.g. “antioxidant”, “free radical scavenger”, “Anti-atherosclerotic”, “cardioprotective”, “nervous system”, “anti-inflammatory”, “anti-cancer”, “signal transduction”, “endothel”, “blood vessel”, “diabetes”, “cell cycle”, “bioactive”, “in vivo”, “clinical”, “preventive”, “therapeutic”) and these results were compiled. The aAuthors also discuss the extraction of polyphenols and their technological potential as food additives in Appendix A and Appendix B, to ease the overview. Finally, some of the pharmaceutical applications of polyphenols are listed using nanobioconjugates such as dendrimers.”
Reviewer’s comment:
Add conclusion and future studies.
Answer: 8.5 Conclusions and future perspectives chapter has added to the text and Summary has removed:
“8.5 Conclusions and future perspectives
Grape polyphenols exert cardioprotective, anti-cancer, anti-diabetic, anti-obesity, anti-osteoarthritis, anti-neurodegenerative and anti-microbial effects both through direct anti-oxidant properties and antioxidant enzyme stimulating effects, and via modulating other signal transducers, for example inducing SIRT-1 gene, and inhibiting NFkappaB and mTOR gene expression, among other inflammatory genes (COX-2, MMPs). Thus, several patented products with high grape polyphenol content for therapeutic application and disease prevention too were developed. The present review article may contribute to further studies, therapeutical approaches and even development of new compounds and products, too.
The physiologicacolly active molecules of the polyphenol fraction can be selectively delivered to diseased cells by binding to dendrimers. Dendrimer binding can increase their chemiluminescence and antioxidant properties. Thus further in vivo and clinical studies are warranted to elucidate their beneficial effects in combination with active pharmaceutical ingredients or food supplements.
The number of studies on the binding of polyphenols to drug carriers, in particular, the formation of nanoscale conjugates, has increased significantly in the last decade. Effective use of polyphenols requires their substitution at higher concentrations than usual. As a major component of grape seed extract, resveratrol plays a central role in the synthesis of anti-cancer polyphenol-containing nanobioconjugates (e.g. dendrimers, polymer nanoparticles, liposomes, nanotubes, micelles, etc.) {Moshawih, 2019 #4846}. Dendrimers, as nanocarriers, can play a prominent role in this, because their multifunctional surface area allows them to contain a high local concentration of active ingredient in a small volume, which makes their application more efficient {Moshawih, 2019 #4846}.
65.1 Chemiluminescent substances
Polyphenol dendrimers are used to enhance chemiluminescence {Nakazono, 2002 #4540}{Sanz del Olmo, 2020 #4560}. Dendrimers made from gallic acid produce singlet oxygen in the presence of hydrogen peroxide. They also have chemiluminescent properties, allowing the presence of singlet oxygen to be detected in chemical systems {Nakazono, 2002 #4540}. Second-generation polyphenol dendrimers were synthesized with various core molecules and chemiluminescence was measured upon reaction with H2 O2 at basic pH. High chemiluminescence was measured for all types of polyphenol dendrimers, which was 120 times higher than that of gallic acid. The intensity of chemiluminescence is strongly dependent on the distance of each branch in the structure of the polyphenol dendrimers {Agawa, 2008 #4542}[175]. Stilbene dendrimers have also been prepared, which also have increased photochemical activity [173, 176, 177]. {Sanz del Olmo, 2020 #4560}{Saberi, 2020 #4558}.
Resveratrol as main component of grape seed extract play a central role in synthesis of polyphenol nanobioconjugates for anticancer drugs
65.2 Increased antioxidant activity
The antioxidant activity of polyphenols from green tea could be significantly enhanced by enzymatic polymerization (polycatechin) or coupling to polyamino-amide dendrimers (PAMAM-catechins) {Kurisawa, 2003 #4546}. First, second, and third generation dendrimers containing two, four, and eight tannic acid groups, respectively, were synthesized. The antioxidant property of the dimer is more than four times that of the monomolecular tannic acid, the tetramer more than twice that of the monomolecular tannic acid, and the octamer one and a half times {Halkes, 2002 #4548}. The antioxidant activity of the vitamin E analogue (tonox) bound to gold nanoparticles is increased eightfold compared to the free molecule {Nie, 2007 #4552}. Naturally occurring polyphenols bind strongly to both proteins and cell membranes. Taking advantage of this, dendrimers with a catechol-modified surface can deliver a wide variety of bioactive proteins and polypeptides into cells. Recent experimental results demonstrate that catechol dendrimers can also kill tumor cells in vivo, e.g. by transporting the enzyme alpha-chymotrypsin into the tumour cell matrix {Zhang, 2022 #4835}.
Because of their general anti-inflammatory and free radical scavenging properties, polyphenols are likely to be widely used as medicines {Gołąbek, 2021 #4827}. The studies presented demonstrate that their therapeutic and preventive role can be significant against the most common pathologies (lung cancer, atherosclerosis, hypertension, diabetes, microbial infections, etc.) {Gupta, 2020 #4829}. The efficacy and/or targeted therapeutic use of polyphenols can also be achieved by using various nanocarriers such as dendrimers {Guo, 2021 #4831}. Active substances (e.g. from grape seed extract), which can be produced cheaply and in an environmentally friendly way on an industrial scale {Oprea, 2022 #4833}, could significantly replace synthetic drugs, which are not only expensive to produce but also are more toxic for the living organisms than polyphenols in general {Moshawih, 2019 #4846}. Combined with the latest therapeutic technologies (e.g. gene therapy), polyphenols could well complement and make the medicine of the future more effective.{Moshawih, 2019 #4846}{Bhaskara, 2020 #4848}.”
Reviewer’s comment:
References
Add available DOI to the reference list.
Answer: DOI identifiers has been added to the reference list.
Reviewer’s comment:
Reference 196: add English names and journal.
Answer: The given reference has removed and a new one has inserted with a full English journal name.
Bestt regards.
On behalf of authors:
Imre Hegedus

Reviewer 5 Report
Dear Authors,
The manuscript presents the review details the antidiabetic, anticarcinogenic, antiviral, vasoprotective, and neuroprotective roles of grape-origin flavonoids.
The article has some scientific value, but there are two main drawbacks.
Firstly, a good review must be informative about the field and focus on a topic in a way that has not been done before. It means that a review article needs to go beyond mere description and ‘state-of-the-literature’ summaries. It should show and develop a new way of thinking about an analyzed topic. In order to enrich the study, the Authors have to mention the current trends in this issue which was missing in the analyzed text.
Secondly, another huge drawback is the very old references (more than 190 references among 241 all references, date back more than 10 years). This is a very significant concern in the case of review articles.
The other suggestions for improvement are as follows:
· Please recheck thoroughly the whole article and improve its grammatical mistakes.
· Please recheck references according to the journal guidelines.
From my standpoint, this manuscript is not appropriate for publication in a reputable journal such as the International Journal of Molecular Sciences, given the above aspects.
Author Response
Answers to Reviewer 5
Dear Reviewer 5,
thank you for your valuable review. Although strict, we fully agree with every sentence, and have therefore substantially rewritten and supplemented our manuscript.
Reviewer’s comment:
Firstly, a good review must be informative about the field and focus on a topic in a way that has not been done before. It means that a review article needs to go beyond mere description and ‘state-of-the-literature’ summaries. It should show and develop a new way of thinking about an analyzed topic. In order to enrich the study, the Authors have to mention the current trends in this issue which was missing in the analyzed text.
Answer: We have added more than 100 new references to literature. In addition, two new tables summarizing in vivo trials and therapeutic trials in the clinical phase have been added. These results are also analyzed in detail in the text, in each case complemented by the molecular mechanism of action of the polyphenol identified. In addition, the latest revolutionary developments in the field of nanobiotechnology involving polyphenols will be discussed. Finally, the manuscript will be concluded with a brief review of the increasing importance of the future of the medicinal applications of vine-derived polyphenolic active substances. Please, find it on the attached reviewed manuscript file.
Reviewer’s comment:
Secondly, another huge drawback is the very old references (more than 190 references among 241 all references, date back more than 10 years). This is a very significant concern in the case of review articles.
Answer: In most places we have deleted old literature or added literature from the last 3 years.
Reviewer’s comment:
- Please recheck thoroughly the whole article and improve its grammatical mistakes.
Answer: grammatical mistakes has rechecked and improved in full text.
Reviewer’s comment:
- Please recheck references according to the journal guidelines.
Answer: References has checked and some mistake has improved.
Reviewer’s comment:
From my standpoint, this manuscript is not appropriate for publication in a reputable journal such as the International Journal of Molecular Sciences, given the above aspects.
Answer: the authors strongly hope that they have managed to improve the quality of their manuscript, which will have a positive impact on the reviewer's opinion.
Best regards.
On behalf of authors:
Imre Hegedus

Round 2
Reviewer 4 Report
This manuscript could be accepted in this present form.
Reviewer 5 Report
Dear Authors,
The majority of previous comments have been taken into account.
I have no more objections to the new version of the manuscript. The Authors improved the paper. They have made a great deal of effort in improving this manuscript.
From my standpoint, this article is appropriate for publication in the International Journal of Molecular Sciences, given the above aspects.